# Function of TREM1 and TREM2 in Liver-Related Diseases

**DOI:** 10.3390/cells9122626

**Published:** 2020-12-07

**Authors:** Huifang Sun, Jianguo Feng, Liling Tang

**Affiliations:** 1Key Laboratory of Biorheological Science and Technology, Ministry of Education, College of Bioengineering, Chongqing University, Chongqing 400044, China; 201819021003@cqu.edu.cn; 2Department of Anesthesiology, The Affiliated Hospital of Southwest Medical University, Luzhou 646000, China

**Keywords:** TREM1, TREM2, inflammation, metabolism, fibrosis, tumorigenesis

## Abstract

TREM1 and TREM2 are members of the triggering receptors expressed on myeloid cells (TREM) family. Both TREM1 and TREM2 are immunoglobulin superfamily receptors. Their main function is to identify foreign antigens and toxic substances, thereby adjusting the inflammatory response. In the liver, TREM1 and TREM2 are expressed on non-parenchymal cells, such as liver sinusoidal endothelial cells, Kupffer cells, and hepatic stellate cells, and cells which infiltrate the liver in response to injury including monocyte-derived macrophages and neutrophils. The function of TREM1 and TREM2 in inflammatory response depends on Toll-like receptor 4. TREM1 mainly augments inflammation during acute inflammation, while TREM2 mainly inhibits chronic inflammation to protect the liver from pathological changes. Chronic inflammation often induces metabolic abnormalities, fibrosis, and tumorigenesis. The above physiological changes lead to liver-related diseases, such as liver injury, nonalcoholic steatohepatitis, hepatic fibrosis, and hepatocellular carcinoma. Here, we review the function of TREM1 and TREM2 in different liver diseases based on inflammation, providing a more comprehensive perspective for the treatment of liver-related diseases.

## 1. Introduction

The liver is an important organ, which is responsible for promoting digestion and metabolism. After nutrients are ingested in the gut, they enter the liver through the portal vein into the blood. There is an abundance of gut-derived molecules in the blood, such as bacterial products, environmental toxins, and food antigens. To maintain homeostasis, the intestinal mucosal barrier will filter these gut-derived molecules. Although the gut barrier is inherently leaky, this leakiness increases after injury, such as alcohol-induced liver injury [1]. Then, a few gut-derived molecules or viruses are able to enter the portal vein and cause damage to the liver [2,3]. Once an infection or tissue damage occurs, homeostasis is broken, and the immune response of the liver is rapidly activated [4].

The triggering receptors expressed on myeloid cells (TREM) family is a receptor family of the immunoglobulin superfamily, which includes five family members: TREM1, TREM2, TREM3, and TREM-like transcript-1 (TLT1) and -2 (TLT2) [5,6]. TREM1 and TREM2 are expressed on the cell membrane where they receive and transmit external signals. Their structure mainly includes a V-type Ig outside the membrane, a short stalk, transmembrane region, and cytoplasmic tails [7]. The crystal structure of the V-type Ig has been used to identify its ligand [8,9,10]. Although the ligands of TREM1 and TREM2 are not well understood, the V-type Ig domain in different cells recognize different ligands, such as HMGB1, Hsp60, lipopolysaccharide, and aminophospholipid [11,12,13,14]. The short stalk is used to connect the V-type Ig and transmembrane region. The transmembrane region is fixed on the cell membrane and transmits the external signal to the cytoplasmic tail. Since the cytoplasmic tail of TREM1 and TREM2 lacks a signaling motif, these proteins rely on DNAX-activation protein of 12 kDa (DAP12) to pass the signal.

In the liver, TREM1 and TREM2 are expressed on non-parenchymal cells (NPCs), such as liver sinusoidal endothelial cells (LSECs), Kupffer cells (KCs), and hepatic stellate cells (HSCs) [15,16,17,18,19,20,21]. Apart from resident liver cells, TREM1 and TREM2 are also expressed on macrophages and neutrophils, which can be recruited to the site of inflammation to participate in innate immune regulation along with the resident immune cells [18,22,23]. In KCs, TREM1 promotes the release of pro-inflammatory cytokines and chemokines, such as interleukin-1β (IL-1β), IL-6, (C-C motif) ligand 2 (CCL2), CCL5, and interferon γ inducible protein-10 (CXCL10) [15,24]. In HSCs, TREM1 develops a pro-inflammatory phenotype by phosphorylating extracellular regulated protein kinases (ERK) 1 [25]. In addition, TREM1 also amplifies endogenous pro-inflammatory signals in monocyte/macrophages [18,26]. TREM2 negatively regulates the synthesis of cytokines, such as tumor necrosis factor-α (TNF-α) and IL-6, in macrophages and dendritic cells [19,27]. TREM2 also inhibits the production of pro-inflammatory factors in KCs and HSCs [20].

It is important to note that inflammation is the basic pathological response of liver diseases. Upon the development of inflammation, it is common to see the development of metabolic abnormalities, fibrosis, and tumorigenesis in disease, such as that seen in non-alcoholic hepatitis, hepatic fibrosis, and hepatocellular carcinoma [28,29,30]. In this review, we evaluate the function of TREM1 and TREM2 in different liver-related biological changes. TREM1 and TREM2 participate in inflammation, metabolism, fibrosis, and tumorigenesis—all of which are closely associated with liver diseases. Based on the function of TREM1 and TREM2 in the liver, they are promising targets for the treatment of liver-related diseases.

## 2. Liver Cell Types

The liver is the biggest digestive gland in the body. It can regulate metabolism and secret bile to maintain normal energy metabolism. Additionally, it takes part in immune responses and the elimination of toxins [31]. Liver tissue consists of hepatocytes and NPCs. Hepatocytes, accounting for 60–80% of all liver cells, take part in detoxification, lipolysis, and gluconeogenesis [32]. Non-parenchymal cells account for about 20–40% of liver cells and include several cell types, including LSECs, KCs, HSCs, and biliary cells [33].

Hepatic lobule is the basic unit of liver structure. There is a central vein in the center of hepatic lobule. Hepatocytes are centered in the central vein and arranged radially to form the hepatic plates. The space between hepatic plates is sinusoids [34]. The sinusoids are special capillaries where blood and hepatocytes exchange substances. Arterial blood from the hepatic artery, which provides oxygen, and venous blood from the hepatic portal vein, which provides nutrients, are mixed in the hepatic sinusoid. LSEC, the principal cell of the hepatic sinusoidal wall, has a unique structural and phenotypic feature [35]. Owing to the lack of a basal lamina and existence of fenestration in LSECs, the hepatic sinusoid has high permeability. Under the protection of LSECs, hepatocytes have access to macromolecules from the blood without the exposure to microbial and food antigens [36]. Furthermore, LSECs take part in innate immune recognition in the liver. Toll-like receptors (TLRs) and scavenger receptors are two kinds of conserved pattern recognition receptors, which are expressed on LSECs and perform different functions [37,38]. When harmful substances invade cells, TLRs recognize these substances and activate downstream signaling pathways. Thus, inflammatory and immune responses are initiated to eliminate invaders. Scavenger receptors determine the endocytic capacity of LSECs. When scavenger receptors recognize extracellular ligands, they internalize the ligand, transport them from the cell membrane into the cytoplasm, and then the lysosome in LSECs clear this “waste” [39]. KCs are a specialized population of macrophages located in the liver sinusoidal vascular space and periportal area. These cells reside on the surface of the LSEC or inset into the intercellular space via pseudopods. Under steady-state conditions, they phagocytize microbial products and other toxic substances delivered from the blood. When KCs are activated, they secret pro-inflammatory cytokines, which recruit immune cells and induce activation of HSC [40]. To respond to changes in environmental signals, KCs are polarized into M1- and M2-type KCs. M1-type KCs mainly promote inflammation, and M2-type KCs suppress inflammation. The balance between M1 and M2 KCs is essential for regulating the occurrence and development of liver damage [41]. There is a cavity between sinus wall and hepatocytes, which is called perisinusoidal space of Disse. HSC is a star-shaped cell, and it is located in the cavity. Under quiescent conditions, storing vitamin A is the main function of HSCs. When HSCs are activated, they promote the fibrotic process, such as the deposition of extracellular matrix. However, these cells also express inflammatory factor genes and participate in the inflammatory response [42,43]. Apart from the liver resident cells in the liver, monocyte-derived macrophages are also recruited to liver-injured sites in response to inflammation [44]. The different intrahepatic macrophage populations, as evidenced by single cell RNA sequencing and immune-related proteins expressed on KCs and HSCs via proteomic analysis, further suggest that both resident immune cells and monocyte-derived macrophages are involved in the innate immune response in the liver [45,46].

There are many antigenic substances in peripheral region, it is very important for LSECs to filter these substances and KCs participate in innate immunity. Inflammation, as the first response of the immune system to external infection, actually plays a protective role. However, excessive acute inflammation or long-term inflammation can also lead to hepatocyte death [47].

## 3. Roles of TREM in Liver Inflammation

### 3.1. Function of TREM in Inflammation

Lipopolysaccharide (LPS), a component of the outer cell wall of Gram-negative bacteria, is an endotoxin. It is the classical pathogen-associated molecular pattern (PAMP) and is often used to induce inflammation. Table 1 shows that the expression of TREM1 and TREM2 can be regulated indirectly by LPS. LPS-induced inflammation is usually mediated by Toll-like receptor 4 (TLR4). When LPS induces inflammation by TLR4, it first combines with CD14 and myeloid differential protein-2 (MD2) to form a dimer. Then, TLR4 regulates the release of downstream inflammatory factors [48]. Although LPS is not the ligand of TREM, the cascade reaction mediated by LPS/TLR4 affects the expression of TREM and, in turn, the initiation of its function.

In acute endotoxemia, LPS induces a rapid upregulation of TREM1, but a downregulation of TREM2 in hepatic macrophages and endothelial cells [18]. TREM1 regulates the TLR4 signaling pathway and expands the inflammatory response by activating myeloid differentiation factor 88 (MyD88), CD14, interleukin 1β (IL-1β), monocyte chemotactic protein 1 (MCP-1), and IL-10 genes [49]. The activated TREM1/TLR4 signaling cascade increases the release of cytokines, such as tumor necrosis factor α (TNF-α), IL-1β, and nuclear binding of transcription factors, such as activator protein 1 (AP-1) and nuclear factor kappa B (NF-κB) [50]. Since the half-life of TREM2 is short and TNF-α inhibits the expression of TREM2, it is reduced in acute inflammation. Because TREM2 is regulated by continuous proteasomal degradation, its turnover rate is high [18,51]. This situation is reversed in chronic liver injury, particularly in mice after being subjected to CCI_4_, a chemically induced liver injury model, for 10 days [52]. In chronic inflammation, KCs are polarized into anti-inflammatory M2-type KCs and protect the liver from chronic inflammation [20,53,54]. In liver ischemia–reperfusion injury, the expression of TREM2 and DAP12 is much higher than other tissues. DAP12 is a transmembrane adaptor shared by TREM1 and TREM2. DAP12 proteins are located on the cytoplasmic side of the cell membrane and form a disulfide-bound homodimer in the cell. DAP12 binds with the receptor chains of TREM1 and TREM2 via a complementary charged transmembrane domain [55]. TREM2/DAP12 negatively regulates inflammation to reduce the liver injury caused by inflammation [19]. Table 1 shows the expression of TREM in different immune-related cells. Both TREM1 and TREM2 mediate downstream signal transduction pathways through DAP12, but TREM1 mainly plays a pro-inflammatory role in acute inflammation. TREM2 primarily suppresses inflammation to perform a protective role in overwhelming inflammation.

### 3.2. Mechanism of TREM in Inflammation

The TLR family includes thirteen proteins, TLR1–TLR13. TLR proteins that have been extensively investigated include TLR2, TLR3, TLR4, TLR5, and TLR9. Both TLR4 and TLR9 are expressed on LSECs, KCs, and HSCs. Since LSECs and KCs are liver resident immune cells, TLR expression is abundant on them. Different TLRs have different ligands. The ligand of TLR3 is double-stranded RNA. LPS, free fatty acids (FFA), heat shock protein 60 (HSP60), and hyaluronan are ligands of TLR4 [61,62,63,64]. The ligand of TLR9 is unmethylated CpG motifs of DNA [65,66].

In the TLR4 signaling pathway, several MyD88 adaptor family members are involved in the activation of TLR4. When TLR4 is activated, MyD88 interacts with TLRs through the help of Toll-IL-1 receptor domains. MyD88 recruits and phosphorylates members of the IL-1 receptor associated kinase (IRAK) family, such as IRAK4 and IRAK1, which then activates the ubiquitin ligase TNF receptor-associated factor 6 (TRAF6). TRAF6 further affects three mitogen-activated protein kinases (MAPK) pathways: p38-MAPK, ERK, and JNK. While both TREM1 and TREM2 are receptors of DAP12, they have different biological function—TREM1 exhibits a pro-inflammatory function and TREM2 exhibits an anti-inflammatory function.

DAP12, the transmembrane immune-adaptor of TREM1 and TREM2, regulates NF-κB via TLR4 activation [67]. When DAP12 responds to the signal transmitted by TREM, its immunoreceptor tyrosine-based activation motif (ITAM) is phosphorylated by Src family tyrosine kinases. Then, the Syk tyrosine kinase and zeta-chain-associated protein kinase 70 kDa (ZAP70) are recruited to regulate the activation of downstream adaptor complexes [68]. Finally, these intermediate adaptor complexes regulate the activation of ERK and the nuclear translocation of NF-κB, leading to the adjustment of the inflammatory response [69,70]. Regardless of promoting inflammation or suppressing inflammation, both TREM1 and TREM2 are dependent on DAP12 and TLR4. When TLR4 is activated by LPS, inflammatory cytokines, such as IL-6, TNF-α, IL-1β, and NF-κB, upregulate the expression of TREM1. TREM1 then augments the TLR4-mediated inflammatory response by binding with DAP12 [57,58]. The pro-inflammatory function is dependent on MAPK and NF-κB signaling pathways, and cytosolic adaptor caspase recruitment domain family member 9 (CARD9) plays an important role in promoting inflammation [71]. The anti-inflammatory function of TREM2 relies on the negatively charged residue of DAP12 in transmembrane region, which interacts with TREM2 via electrostatic interaction [72]. This interaction results in partial activation of DAP12, resulting in the inhibitory signal [73]. In addition, the inhibitory effect of TREM2 is also achieved by the A20 negative regulator of TLR4 [51]. TREM2 reduces the level of inflammation by reducing the activation proteins downstream of TLR4, such as p38-MAPK and ERK [20]. Figure 1 shows the inflammatory regulation mechanism in a TREM1/2-TLR4-dependent manner in the liver.

The mechanism of expression level reversal between TREM1 and TREM2 has yet to be fully investigated in the liver. However, based on the change in TREM1 and TREM2 expression level, we can propose that the expression of TREM2 is injury dependent. When the level of inflammation is too high, TREM2 may function to reduce excessive immune injury, in a manner similar to the polarization of macrophages from M1 to M2 during the wound healing process. Although this conjecture remains to be verified, it is a good direction for researchers to start investigating a probable alterative mechanism, as well as its use for future liver inflammation treatment.

## 4. Roles of TREM in Metabolism

Fatty liver is a multifactor-mediated pathological change of hepatocytes, which is not only caused by fat accumulation, but also related to genetics, environment, and metabolic stress. Thus far, the mechanism by which simple steatosis develops into non-alcoholic steatohepatitis (NASH) is unclear, but damage such as inflammation, lipotoxicity, insulin resistance, and oxidative stress promotes NASH [74]. Moreover, NASH may progress into cirrhosis and hepatocellular carcinoma [75,76,77,78]. When lipid over-accumulates in the liver, the lipid droplets causes lipotoxicity to hepatocytes by releasing free fatty acids [79]. In addition, lipotoxic hepatocytes attract inflammatory cells, which activate the inflammatory response [80]. During the early stage of steatohepatitis, toxic lipid metabolites accumulate in fat-laden KCs, which produces high levels of pro-inflammatory cytokines or chemokines [81]. Many natural killer T cells are distributed in the visceral adipose tissue to participate in innate immunity via secreting a series of cytokines and causing cytotoxic effects [82]. The lipid released from dead hepatocytes are taken up by activated KCs to reduce liver lipid accumulation [83]. In addition, M2-type macrophages can be polarized to M1-type macrophages by fatty acids, which promote inflammation and ectopic lipid accumulation [84]. Other macrophage subsets are believed to be involved in NASH and have a unique lipid metabolism [85]. Although adipose tissues contain many different kinds of cells, the information about TREM1 and TREM2 has not been well investigated. Current research indicates that both TREM1 and TREM2 mainly express on immune cells. These immune cells are the natural killer T cells, macrophages, and KCs mentioned above.

In NASH, the severity of fatty liver is proportional to the expression of TREM1 and inversely proportional to the expression of TREM2 [86]. Expression of TREM1 is upregulated, and overexpression of TREM1 increases lipid accumulation-related genes via the PI3K/AKT signaling pathway in non-alcoholic fatty liver disease (NAFLD) [87]. A previous study demonstrated that TREM2 recognizes glycerophospholipids and sphingomyelins. In particular, the recognition of anionic lipids by TREM2 in an Alzheimer’s disease model has attracted extensive attention [88,89]. Recently, NASH-associated macrophages (NAMs) have been found in a NASH mouse model. NAMs not only have a high expression of TREM2, but also have enriched expression of endocytosis and lysosomal degradation genes [90]. Upon further research, lipid-associated macrophages (LAMs) were found. To reduce the occurrence of adipocyte hypertrophy and systemic metabolic dysregulation, LAMs adjust lipid uptake and metabolism by a TREM2-dependent signaling pathway. Of note, the metabolic character of LAMs is conservative in the liver, and the immunosuppressive genes, Lgals1 and Lgals3, were expressed [91]. TREM2-mediated inflammation is also involved in lipid accumulation. TREM2 affects lipid peroxidation by changing metabolic pathways and reactive oxygen species (ROS) production in liver macrophages after liver injury [20]. Thus, we conclude that TREM1 not only promotes inflammation, but also promotes lipid accumulation. However, TREM2 reduces the excessive accumulation of lipids by phagocytosis and suppressing lipid peroxidation and ROS.

## 5. Roles of TREM in Hepatic Fibrosis

Fibrosis occurs in multiple organs, which is represented as the abnormal hyperplasia of connective tissue. Many factors lead to hepatic fibrosis, such as viral hepatitis, alcohol abuse, and drugs [92,93,94]. Inflammation and fibrosis are two processes that are activated in response to injury as reparative mechanisms. However, when there is chronic liver injury, chronic inflammation and fibrosis lead to pathological conditions [29,95]. HSCs play an important role in fibrosis process, which can be activated by IL-6, transforming growth factor-β (TGF-β), and platelet-derived growth factor [96,97,98]. After liver injury, KCs not only produce pro-inflammatory factors to activate HSCs but also recruit CCR2+/Ly-6Chi monocytes into injured liver to sustain activation of HSCs [99]. After HSCs are activated, they transform into myofibroblasts. The most obvious biological change caused by this transformation is the deposition of extracellular matrix (ECM). The components of excessive matrix deposition mostly are type I collagen, type III collagen, and type V collagen. In liver fibrosis, type I collagen and type III collagen are relatively increased, which makes ECM become denser [100].

In a healthy liver, the expression of TREM1 is limited. In the process of hepatic fibrosis, the expression of TREM1 in KCs is significantly increased and inflammatory factors are released. Then, these inflammatory factors will induce the activation of HSCs [58]. When TREM1 is inhibited, the TREM1-TLR2/4-MyD88-mediated inflammatory signaling pathway is inhibited, and the polarization of M1-type macrophages are also reduced [101]. TREM2 expressed on KCs and HSCs dampen TLR4-induced inflammation, and the combined action of liver resident and infiltrating cells is also necessary to dampen inflammation [20]. A decrease in inflammation allows for wound healing and restores tissue homeostasis (Figure 2). However, a recent study highlighted a new role of TREM2 in hepatic fibrosis, which differs from its protective role in inflammation. Through single-cell RNA sequencing, a scar-associated subpopulation of macrophages has been found, which is identified by TREM2^+^ and CD9^+^ in human and mouse fibrotic livers. These cells also have the characteristics of monocytes and KCs. However, the function of these cells is pro-fibrotic, and they work with endothelial cells and mesenchymal cells to promote fibrosis through TNFRSF12A, PDGFR, and NOTCH signaling pathways [102].

Tissue homeostasis is maintained by a balance between injury and wound healing. Wound healing is an important biological process in the injured liver, especially in liver fibrosis. In the context of chronic inflammation, clearance of activated HSCs may be achieved by reducing the level of inflammation. Inhibition of NF-κB induces the apoptosis of HSCs [103,104]. Inhibition of CCR2 and CCR5 also suppresses the migration and fibrogenic potential of HSCs [105]. TGF-β released from KCs activates HSCs and promotes fibrosis [106]. M1 macrophages promote fibrosis by producing TGF-β, and M2 macrophages can also be pro-fibrogenic by secreting TGF-β during chronic inflammation [99,107]. Inhibition of the TGF-β signaling pathway of HSCs significantly improves collagen deposition [108]. Chronic inflammation is often the cause of liver fibrosis. In this case, there are two main ways to decrease excessive wound healing. One approach is to reduce NF-κB, which promotes the apoptosis of activated HSCs and reduces fibrosis. The other approach is to reduce the level of TGF-β released from macrophages and KCs, which reduces the collagen deposition to dampen fibrosis. During acute inflammation, TREM1 expressed on resident NPCs and macrophages induce inflammatory factors that remove the antigen. However, HSCs are also activated to promote scar formation and tissue repair. During chronic inflammation, persistent activation of HSCs leads to liver fibrosis. To reverse fibrosis, we can reduce the level of inflammation by regulating TREM1 and promoting HSC activity reversion and apoptosis.

## 6. Roles of TREM in Liver Tumorigenesis

Hepatocellular carcinoma (HCC) is a common liver-related disease, but it has a high mortality rate [109]. Major causes of HCC include long-term viral infections, non-alcohol-associated liver disease or alcoholic injury, environmental carcinogens, and hereditary genetic disorders [110,111,112]. Abnormal proliferation and migration of cancer cells are the typical features of the high lethality of HCC. In liver tissue, hepatocytes can self-repair the liver through proliferation. However, when the newborn hepatocytes are exposed in chronic inflammation, the normal liver regeneration reaction will be transformed into excessive proliferation, accompanied by gene change, which will eventually lead to liver tumorigenesis [113].

TREM1 can be found in most tumor samples, whether it is in tumor tissue or adjacent tissue. TREM1 expressed on KCs promotes inflammation by increasing the expression of IL-6, IL-1β, and CCL2 [15]. In HCC, the activated HSCs expressing TREM1 promote the migration of HCC cells via soluble TREM1 at the cellular level [17]. TREM1 also regulates phosphorylation of STAT3, AKT, ERK, and p65 to promote the proliferation of HCC cells [114]. From the perspective of the tumor microenvironment, TREM1^+^ tumor-associated macrophages (TAMs) are abundant in the hypoxic environment of hepatocellular carcinoma. When TREM1 expression was blocked, the dysfunction of CD8^+^ T cells was markedly attenuated. In a hypoxic environment, hypoxia inducible factor 1α induces the expression of TREM1 on TAMs and eventually leads to immunosuppression by indirectly weakening the cytotoxic functions of CD8^+^ T cells and inducing CD8^+^ T cells apoptosis. Although the expression of programmed cell death ligand 1 (PD-L1) was high in TAMs, TREM1 and PD-L1 expression is not related. In the process of promoting tumor formation, the role of TREM1 is based on the dysfunction of CD8^+^ T cell. Considering that CD8^+^ T cells play a critical role in anti-PD-L1 drug therapy, this result suggests that TREM1^+^ TAMs can be a potential target in anti-PD-L1-mediated immunotherapy in HCC [115,116,117].

TREM2 was first studied in the liver cancer cell lines MHCC97H and HepG2, and the expression level was high. To investigate the function of TREM2 in liver cancer, TREM2 was knocked down by siRNA. Then, it inhibited cell proliferation, induced cell apoptosis, and blocked cell cycle in vivo, which suggested that TREM2 was an oncogene in the tumorigenesis [118]. In a study using HepG2 cells and an HCC model animal, PI3K/AKT was the pathway induced by TREM2 to regulate the biological behavior of the tumor cells [119]. In the PI3K/AKT signaling pathway, TREM2 activates PI3K and phosphorylates AKT. AKT then participates in multiple cellular processes, such as proliferation, apoptosis, and metastasis. After AKT is phosphorylated, it mediates the phosphorylation of downstream molecules, such as p21, a cell cycle inhibitor. Once phosphorylated, p21 is degraded and downregulates expression of pro-apoptotic protein Bax, promoting the apoptosis of tumor cell [120,121]. In the above studies, we conclude that the role of TREM2 in HCC is to promote tumorigenesis. However, a recent study provides a different view. In this study, TREM2 overexpression suppressed HCC progression by regulating the PI3K/Akt/β-catenin pathway. Furthermore, reduced TREM2 expression is observed in the poor prognosis of HCC patients [122]. Another study supports the protective role of TREM2 in hepatocarcinogenesis. TREM2 attenuates inflammation, oxidative stress and Wnt ligand secretion, preventing hepatocyte proliferation and damage [21]. The differing functions of TREM2 in vivo and in vitro may be caused by differences in animals and cell lines [123].

Aside from HCC, TREM2 has also been studied in many other tumors, such as colorectal carcinoma, lung cancer, and glioma [124,125,126,127,128]. However, it remains to be explored whether the mechanism of TREM2 in liver tumors and other tumors is consistent. Owning to the complexity of HCC, there are still many issues regarding the function and mechanism of TREM1 and TREM2 that need to be further studied, including the role of the gene and the tumor microenvironment.

## 7. Conclusions

Most liver diseases are chronic and often accompanied by inflammation. An inflammatory response is a common clinical pathological process and exists in different tissues and organs. Chronic inflammation in the liver changes many biological characteristics, resulting in metabolic abnormalities, fibrosis, and tumorigenesis. Therefore, inflammation plays an important role in the occurrence and development of liver diseases. TREM1 and TREM2 are immunoglobulin superfamily receptors, which often regulate innate immunity via the inflammatory response. Importantly, the NPCs of the liver express TREM1 and TREM2 to regulate the innate immune response. Based on the inflammatory regulation of TREM, these resident immune cells can interact with the infiltrating immune cells during hepatic inflammation to regulate the metabolism, fibrosis, and tumor diseases in the liver. The biological function mediated by TREM1 and TREM2 is shown in Figure 3.

In this review, we evaluate the function of TREM1 and TREM2 in liver injury, NASH, hepatic fibrosis, and HCC. TREM1 plays a disease-promoting role via amplifying inflammation, promoting lipid accumulation, hepatic fibrosis, and accelerating tumor progress. TREM2 plays a protective role by suppressing inflammation, decreasing lipid-related gene expression, and suppressing tumor progression. However, the pro-fibrotic TREM2^+^CD9^+^ subpopulation of macrophages makes the function of TREM2 in the liver much more complex. Although the function of TREM2^+^CD9^+^ macrophages in fibrosis is different, it deserves to be further investigated. Thus far, studies regarding the use of TREM1 and TREM2 in therapies for liver disease therapy are still at the experimental level. Nonetheless, TREM proteins have the potential to be used as a therapeutic target based on the mechanism of TREM in different stages of liver diseases. It is necessary to further study TREM1 and TREM2 in the liver to advance the treatment of liver-related diseases.

## Figures and Tables

**Figure 1 cells-09-02626-f001:**
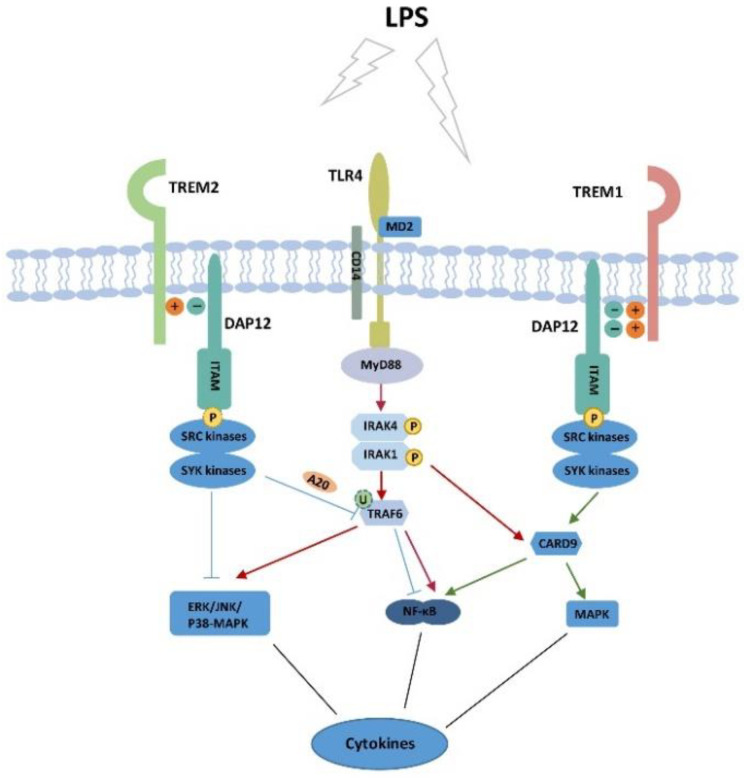
The inflammatory regulation mechanism in a TREM1/2-TLR4-dependent manner in the liver. Upon the stimulation LPS, TLR4 is activated and produces cytokines via NF-κB and MAPK signaling pathways. TREM1 phosphorylates ITAM by activating Src and Syk kinases and regulates CARD9. The upstream signal passed to CARD9 augments inflammation via NF-κB and MAPK. The IRAK1 and TRAF6 in TLR4 signaling pathway also promote inflammation via CARD9 and NF-κB, respectively. TREM2 mainly alleviates the inflammation by binding to DAP12 with low affinity. Partial activation of DAP12 reduces the phosphorylation Src and Syk kinases, inhibiting the inflammatory factor mediated by ERK/p38-MAPK/JNK pathway. TREM2 also makes A20 negatively regulate ubiquitination of TRAF6 and decreases inflammation mediated by NF-κB. LPS, lipopolysaccharide; TREM2, triggering receptors expressed on myeloid cells 2; DAP12, DNAX-activation protein of 12 kDa; TLR4, toll-like receptor 4; MD2, myeloid differential protein-2; IRAK, IL-1 receptor associated kinase; TRAF6, TNF receptor associated factor 6; ITAM, immunoreceptor tyrosine-based activation motif; Syk, spleen tyrosine kinase; CARD9, cytosolic adaptor caspase recruitment domain family member 9; MyD88, myeloid differentiation factor 88; MAPK, mitogen-activated protein kinases pathway; ERK, extracellular regulated protein kinases pathway; P, phosphorylation; U, ubiquitination.

**Figure 2 cells-09-02626-f002:**
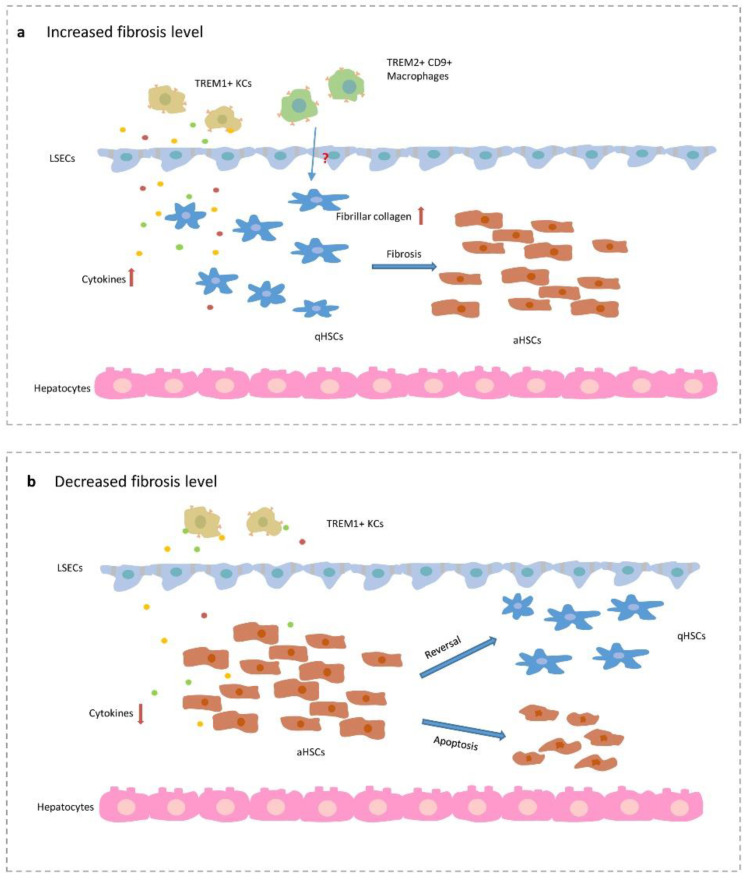
The function of TREM1 and TREM2 in fibrosis based on inflammation. (**a**) In pro-fibrogenic process, TREM1 expressed on Kupffer cells promotes the activation of hepatic stellate cells by increasing cytokines. TREM2^+^CD9^+^ macrophages also upregulate fibrillar collagen expression. (**b**) The decreased expression of TREM1 in KCs reduces cytokines and weakens the activation of quiescent hepatic stellate cells. In the low inflammatory level, activated hepatic stellate cells reverse into quiescent state and apoptosis. TREM, triggering receptors expressed on myeloid cells; LSECs, liver sinusoidal endothelial cells; aHSCs, activated hepatic stellate cells; qHSCs, quiescent hepatic stellate cells; KCs, Kupffer cells.

**Figure 3 cells-09-02626-f003:**
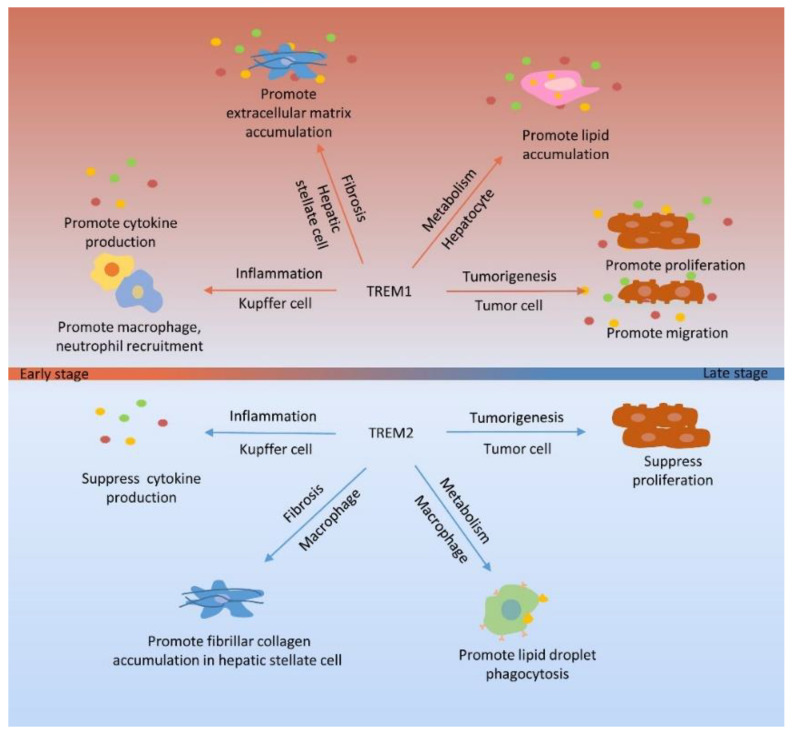
The biological process of TREM1 and TREM2 involved in liver-related diseases. In the early stage of liver-related diseases, the biological function of TREM1 is dominant. To maintain homeostasis, the biological function of TREM2 gradually occupies a dominant position in the late stage. In the inflammatory response, TREM1 expressed on Kupffer cells upregulates production of cytokines and recruits macrophages and neutrophils, which augments inflammation. However, TREM2 expressed on Kupffer cells suppresses production of cytokines. In fibrosis process, pro-inflammatory response of Kupffer cells induced by TREM1 activates hepatic stellate cells. TREM1 also promotes the deposition of extracellular matrix in hepatic stellate cells. TREM2 expressed on macrophages promotes fibrillar collagen accumulation in hepatic stellate cells. In metabolic regulation, TREM1 expressed on hepatocytes promotes lipid accumulation, which causes lipotoxicity to hepatocytes. TREM2 expressed on macrophages increases the phagocytosis of lipid droplet. In tumorigenesis process, TREM1 expressed on tumor cells promotes proliferation and migration, but TREM2 is to suppress proliferation of tumor cells. TREM, triggering receptors expressed on myeloid cells.

**Table 1 cells-09-02626-t001:** The expression of TREM1 and TREM2 in liver-related immune cells.

Name	Cell Type	Cell/Animal	Expression Change	Biological Function	Induction Time	Involved Cytokine	Induction Method	Involved Pathway	Ref
**TREM1**	Human monocyte	Cell	Up	Promote inflammation	24 h	IL-1β, TNF-α, IL-8	LPS	/	[56]
	Rat macrophage	Cell	Up	Promote inflammation	24/48 h	TNF-α, IL-8	LPS	NF-κB	[57]
	Liver macrophage	Animal, TLR4-mutant	Up	/	3 h	TNF-α, IL-1β	LPS	TLR4	[18]
	Liver endothelial cell	Animal, TLR4-mutant	Up	/	3 h	TNF-α, IL-1β	LPS	TLR4	
	Hepatic stellate cell	Animal, TREM1^−/−^	/	Promote fibrosis	6 w	ACTA2, MMP10, TIMP1	CCL_4_	/	[58]
	Kupffer cell	Animal, TREM1^−/−^	/	Promote inflammation	6 w	CCL9, CXCL2, TGF-β	CCL_4_	/	
	Neutrophil	Alcoholic liver disease model	/	Promote inflammation	5 w	MCP-1, TNF-α, IL-1β,	Lieber–DeCarli liquid diet	TLR	[59]
**TREM2**	Bone marrow macrophage	Animal, TREM2^−/−^	/	Suppress inflammation	24 h	IL-6 IL-1β	LPS	TLR4	[60]
	Peritoneal macrophage	Animal, TREM2^−/−^	/	Suppress inflammation	6 h	IL-6, TNF-α	LPS	TLR4	[51]
		Cell	Down	/	1 h		LPS		
	Liver macrophage	Animal, TLR4-mutant	Down	/	3 h	TNF-α, IL-1β	LPS	TLR4	[18]
	Liver endothelial cell	Animal, TLR4-mutant	Down	/	3 h	TNF-α, IL-1β	LPS	TLR4	
	Kupffer cell	TREM2^−/−^	/	Suppress infection	40 h	/	Malaria parasite	/	[16]
	Hepatic stellate cell	Cell	Up	Suppress inflammation	8 w	/	CCL_4_	TLR4	[20]
		Animal, TREM2^−/−^	/	Suppress inflammation	8 w	/	CCL_4_	TLR4	

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
