# Peer review of "Function of TREM1 and TREM2 in Liver-Related Diseases"

_cells, 2020, doi:10.3390/cells9122626_

Round 1

Reviewer 1 Report

Reviewer Comments to Author:

This manuscript by Sun H, Feng J and Tang L is a revision of a previously submitted review manuscript about the role of TREM1 and TREM2 in liver disease. Although authors have tried to address some of the concerns raised in the first submission, there are still many conceptual errors from the scientific point of view as well as vague sentences that could lead to a misleading understanding of the potential role of these receptors in liver diseases. Still, there are many sentences in this manuscript that lack accuracy and are not appropriate from a scientific point of view.

Major Comments

- Starting from the abstract there are concepts that are not scientifically correct. “In the liver TREM1 and TREM2 are expressed on non-parenchymal cells, macrophages and neutrophils.” Non-parenchymal cells include liver resident cells or Kupffer cells. Therefore, either authors include in brackets some examples of non-parenchymal cells and then add other cells such as recruited immune cells including monocyte-derived macrophages and neutrophils or explain which are the type of macrophages they are mentioning in that particular sentence.

- The sections are not well-organized. One example includes the first section of the manuscript (1. Introduction), in which both the structure of TREM family of receptors is described as well as the role of TREM1 and TREM2 in different cell types and diseases, ending with the pathophysiology of the liver. There are no links between the different concepts and ideas are just written without any connection.   

- When describing the liver cell types there are also two sentences that are misleading. “Most NPCs are immune cell, and they participate in metabolism and the fibrotic process. Thus, NPCs are very important in the pathological process of liver diseases.” NPCs are not important for their contribution to metabolism and fibrosis but for their contribution in the inflammatory process. Both inflammation and fibrosis are two processes that are activated in response to injury as reparative mechanisms as a physiological mechanism. However, when there is chronic liver injury, chronic inflammation and fibrosis can lead to pathological conditions. These basic aspects are not even clearly defined. 

- Figure 1 is not correct. TREM2 does not seem to regulate NFkB pathway. In the image blue lines seem to suppress the inflammation downstream of TREM2 negatively regulating MAPK and ERK. ERK kinase is a MAPK, so what do the authors mean with MAPK and ERK? Downstream of MAPKs is not NFfB, what do they mean with NFkB being in the nuclei? The Figure does not show what is known about the crosstalk between TREM2 and TLR4 signaling pathways. Moreover, it is well-known that one of the most obvious differences between TREM1- and TREM2- activated downstream signaling pathways upon TLR4 activation is the difference in NFkB regulation. In this sense, a better approach in this case would have been to include TREM1 and TREM2 signalling cascades upon TL4 activation in the same figure.

- Figure 2 is also not correct. First of all, TREM1 and TREM2 are not present in the figure and secondly, the information included in the figure legend is incorrect and misleading. Part b of the figure says “In chronic inflammation, TREM2 expressed on Kupffer cells and macrophages suppress inflammation and decrease fibrosis by promoting the reversal and apoptosis of activated hepatic stellate cells.” As mentioned by the authors, a recent manuscript published in Nature by Ramachandran et al in 2019, has identified a new population of scar-associated macrophages expressing TREM2+ and CD9+ in human and mouse fibrotic livers, with pro-fibrogenic functions. Therefore, the role of TREM2 regulating the fibrogenic process is not as authors state, decreasing fibrosis.

- Similarly, Figure 3 does not add any relevant information about TREM1 and TREM2 in each condition.

- The role of TREM1 and TREM2 in liver cancer is also not properly described and cited. Regarding the role of TREM1 in HCC it is mentioned that “TREM1 expressed on KCs also promotes inflammation by increasing the expression of IL-6, IL-1β, and CCL2.” Authors should include here PMID: 22719066. Similarly there is a recent manuscript about the role of TREM2 in HCC that is very relevant that was not included in this manuscript that was published in Gut (PMID: 32907830).

Overall, there are many aspects in this manuscript that need to be very cautiously revised, as there are many statements that are misleading and/or incorrect. Moreover, the ideas are not properly organized and Figures are not correct.

Author Response

Response to Reviewer 1:

Major comments

Comment 1: Starting from the abstract there are concepts that are not scientifically correct. “In the liver TREM1 and TREM2 are expressed on non-parenchymal cells, macrophages and neutrophils.” Non-parenchymal cells include liver resident cells or Kupffer cells. Therefore, either authors include in brackets some examples of non-parenchymal cells and then add other cells such as recruited immune cells including monocyte-derived macrophages and neutrophils or explain which are the type of macrophages they are mentioning in that particular sentence.

Answer: We have rewritten this sentence in line 16-18.

Comment 2: The sections are not well-organized. One example includes the first section of the manuscript (1. Introduction), in which both the structure of TREM family of receptors is described as well as the role of TREM1 and TREM2 in different cell types and diseases, ending with the pathophysiology of the liver. There are no links between the different concepts and ideas are just written without any connection.

Answer: We have re-organized this section and deleted redundancy.

Comment 3: When describing the liver cell types there are also two sentences that are misleading. “Most NPCs are immune cell, and they participate in metabolism and the fibrotic process. Thus, NPCs are very important in the pathological process of liver diseases.” NPCs are not important for their contribution to metabolism and fibrosis but for their contribution in the inflammatory process. Both inflammation and fibrosis are two processes that are activated in response to injury as reparative mechanisms as a physiological mechanism. However, when there is chronic liver injury, chronic inflammation and fibrosis can lead to pathological conditions. These basic aspects are not even clearly defined.

Answer: We have deleted these misleading sentences and defined the pathological change of chronic inflammation and chronic fibrosis in line 245-248.

Comment 4: Figure 1 is not correct. TREM2 does not seem to regulate NFkB pathway. In the image blue lines seem to suppress the inflammation downstream of TREM2 negatively regulating MAPK and ERK. ERK kinase is a MAPK, so what do the authors mean with MAPK and ERK? Downstream of MAPKs is not NFfB, what do they mean with NFkB being in the nuclei? The Figure does not show what is known about the crosstalk between TREM2 and TLR4 signaling pathways. Moreover, it is well-known that one of the most obvious differences between TREM1- and TREM2- activated downstream signaling pathways upon TLR4 activation is the difference in NFkB regulation. In this sense, a better approach in this case would have been to include TREM1 and TREM2 signaling cascades upon TL4 activation in the same figure.

Answer: We have revised Figure 1 and added TREM1 mediated TLR4 signaling pathway. More information about the difference between TREM1 and TREM2 is given.

Comment 5: Figure 2 is also not correct. First of all, TREM1 and TREM2 are not present in the figure and secondly, the information included in the figure legend is incorrect and misleading. Part b of the figure says “In chronic inflammation, TREM2 expressed on Kupffer cells and macrophages suppress inflammation and decrease fibrosis by promoting the reversal and apoptosis of activated hepatic stellate cells.” As mentioned by the authors, a recent manuscript published in Nature by Ramachandran et al in 2019, has identified a new population of scar-associated macrophages expressing TREM2+ and CD9+ in human and mouse fibrotic livers, with pro-fibrogenic functions. Therefore, the role of TREM2 regulating the fibrogenic process is not as authors state, decreasing fibrosis.

Answer: We have corrected Figure 2.

Comment 6: Similarly, Figure 3 does not add any relevant information about TREM1 and TREM2 in each condition.

Answer: We have revised Figure 3 and given detailed information about the regulation of TREM1 and TREM2 in inflammation, metabolism, fibrosis and tumorigenesis.

Comment 7: The role of TREM1 and TREM2 in liver cancer is also not properly described and cited. Regarding the role of TREM1 in HCC it is mentioned that “TREM1 expressed on KCs also promotes inflammation by increasing the expression of IL-6, IL-1β, and CCL2.” Authors should include here PMID: 22719066. Similarly there is a recent manuscript about the role of TREM2 in HCC that is very relevant that was not included in this manuscript that was published in Gut (PMID: 32907830).

Answer: We have corrected the citation and added the manuscript about TREM2 in line 335-337.

Reviewer 2 Report

This is a revision of a previously submitted manuscript by Sun et al. which strives to summarize available data on how TREM1 and TREM2 regulate inflammation within the context of different liver diseases (metabolic liver disease, fibrosis and hepatocellular carcinoma). This manuscript benefited from a significant revision of English grammar and usage, but it would still benefit from additional clarity and attention to logical flow and progression of ideas.

Outlined below are some suggestions for further revision

Major comments.

  • I think it is inappropriate to state the chronic liver injury is the only thing that causes hepatocyte death (for example see lines 74 and X). Several examples of acute liver injury which cause direct cytotoxic effects on hepatocytes exists. For example, acetaminophen overdose is an acute injury which causes significant hepatocyte death by necrosis.
  • Beginning around line 61. The authors should note that the gut barrier is inherently leaky, and this leakiness is increased in the context of disease (i.e. alcohol-associated liver disease). This inherent leakiness is why Kupffer cells (KC) are found in higher numbers in the periportal regions of the liver. However, they are inherently less likely to respond to the various gut-derived antigens until something changes. In the context of chronic alcohol consumption, increased gut permeability coupled to increased KC sensitivity to lipopolysaccharide (LPS) results in inflammation which precipitates inflammation associated with alcohol-associated liver disease. Finally, I think the use of ‘toxins’ is not the best word choice. Certainly, molecules liberated through digestion and delivered to the liver are not inherently toxic.  Recommend using a phrase such as ‘gut derived molecules’ and listing bacterial, viral, dietary, environmental antigens.
  • There is significant redundancy in the manuscript. For example, under the subheading ‘liver cell types’ there is a discussion of M1 vs M2 macrophages in lines 107 – 109 and again in the same paragraph, lines 117 – 121. The discussion of M1 vs M2 macrophages should all occur in the same section of the manuscript to improve flow of ideas. The same goes for describing TREM1 and TREM2 in inflammation.
  • Line 122: the first sentence should have a citation. The second sentence does not adequately support the first sentence I would anticipate was intended. For example, while NPC regulation of metabolism would help HC maintain HC function, how does inflammation and liver toxicity do this? I can see a picture where NPC can help HC secrete proteins important for homeostasis. Furthermore, the rest of this paragraph does not relate to the topic sentence making the purpose of the paragraph confusing. It seems to me the purpose of this paragraph was to talk about the ways in which NPC help regulate normal hepatic function while the subsequent paragraphs discuss how NPC, regulated by TREM1 and TREM2, regulate pathogenic processes.
  • Figure 1: if TREM2 negatively regulates LPS signaling, why are arrows pointing to MAPK and ERK? One would anticipate that a symbol indicating a blockade would be more appropriate. Also, the figure could be improved by including TREM1 signaling so that the reader can compare and contrast the effects of TREM1 and TREM2 on TLR4 signaling. Finally, the authors need to make is much more clear how TREM2 inhibits TLR4 signaling – it is counterintuitive that kinase activation would inhibit TLR4-mediated signaling. If the precise mechanism by which TREM2 negatively regulates TLR4 signaling is unknown, then state this, otherwise, provide the mechanistic data to clarify this important point. For example, and based on a information suggesting that low avidity binding of ligands to TMEM2 induces SHP phosphatase, is this a/the mechanism by which TREM2 can negatively regulate TLR4 signaling (reference 84)?
  • Beginning in line 218 (TREM in metabolism) there is considerable data regarding now macrophages regulate adipose tissue. Why do the authors only focus on NK cells? A more balanced approach would be to include both macrophages and NK cells. If there is no information on TREM1 or 2 in adipose, then state this is an excellent area for future investigation.
  • There is a lot of discussion of TREM1 and TREM2 in non-hepatic organs or diseases which occur in the sections describing TREM1 and TREM2 in liver. All discussion of non-hepatic TREM function should be described before diving into liver-specific TREM function (i.e. introduction)
  • In the fibrosis section, please remove redundancy and explain fibrogenesis in a logical manner. For example, inflammation usually precedes fibrosis, yet inflammation is mentioned in the last sentence of this section’s introductory paragraph.
  • Figure 2 needs additional labels and or legends. What are the little colored bars supposed to represent? The legend needs revision for English grammar and usage.
  • Beginning in line 310: this paragraph needs to be re-written for clarity. The thoughts need to be reorganized to tell a more logical story. For example, it would make more sense to describe immune impairments, then mechanism, then how TREM is involved.

Minor comments.

  • Abstract starting line 15: The authors state non-parenchymal cells (NPC) but then list macrophages and neutrophils which is somewhat confusing. For example, Kupffer cells (KC) are resident hepatic macrophages. This sentence could be improved by saying “…NPC and cells which infiltrate the liver in response to injury including monocyte-derived macrophages and neutrophils.” (or equivalent)
  • Abstract starting line 17: I think this sentence on TREM1 and TREM2 would be improved if you indicate that both receptors depend on TLR4 as stated in the review.
  • Line 38: correct the spelling of ‘lipopolysaccharide’
  • Line 88: it is more typical to call the specialized hepatic vessels ‘sinusoids’ instead of ‘sinuses’. Also, here it would be informative to state the location at which this blood enters the liver.
  • Line 109: somewhere in the discussion of HSC, the authors should state ‘perisinusoidal space of Disse’
  • Line 126: You should state something like ‘infiltrating macrophages’ or ‘moncyte-derived macrophages’ to distinguish these macrophages from KC which are also macrophages.
  • Beginning line 131: how does upregulation of TREM1 increase macrophage and neutrophil recruitment? You state TLR4, but, presumably, this is mediated by macrophage and neutrophil-tropic chemokines, respectively, induced through TLR4 signaling. Can you be more specific behind the mechanisms involved?
  • Line 154: do you mean ‘because’ instead of ‘although’ to start this sentence?
  • Line 251: please include a reference for the last sentence in this paragraph.
  • Line 288: note that acute inflammatory responses can also cause hepatocyte death.
  • Line 283: instead of ‘fibrosis’ do you mean ‘fibrogenic potential’ of HSC?
  • Line 312: do you have a reference backing this assertion?
  • The legend to figure 3 needs more explanation. Moreover, it might be relevant to think about adding blocking arrows to show how TREM1 and TREM2 counterbalance one another.
  • Line 367: tumor progression (and not ‘progress’?)

Author Response

Response to Reviewer 2:

Major comments

Comment 1: I think it is inappropriate to state the chronic liver injury is the only thing that causes hepatocyte death (for example see lines 74 and X). Several examples of acute liver injury which cause direct cytotoxic effects on hepatocytes exists. For example, acetaminophen overdose is an acute injury which causes significant hepatocyte death by necrosis.

Answer: We have deleted this sentence.

Comment 2: Beginning around line 61. The authors should note that the gut barrier is inherently leaky, and this leakiness is increased in the context of disease (i.e. alcohol-associated liver disease). This inherent leakiness is why Kupffer cells (KC) are found in higher numbers in the periportal regions of the liver. However, they are inherently less likely to respond to the various gut-derived antigens until something changes. In the context of chronic alcohol consumption, increased gut permeability coupled to increased KC sensitivity to lipopolysaccharide (LPS) results in inflammation which precipitates inflammation associated with alcohol-associated liver disease. Finally, I think the use of ‘toxins’ is not the best word choice. Certainly, molecules liberated through digestion and delivered to the liver are not inherently toxic.  Recommend using a phrase such as ‘gut derived molecules’ and listing bacterial, viral, dietary, environmental antigens.

Answer: We have added related information and corrected this word in line 33-36.

Comment 3: There is significant redundancy in the manuscript. For example, under the subheading ‘liver cell types’ there is a discussion of M1 vs M2 macrophages in lines 107 – 109 and again in the same paragraph, lines 117 – 121. The discussion of M1 vs M2 macrophages should all occur in the same section of the manuscript to improve flow of ideas. The same goes for describing TREM1 and TREM2 in inflammation.

Answer: We have deleted the redundancy.

Comment 4: Line 122: the first sentence should have a citation. The second sentence does not adequately support the first sentence I would anticipate was intended. For example, while NPC regulation of metabolism would help HC maintain HC function, how does inflammation and liver toxicity do this? I can see a picture where NPC can help HC secrete proteins important for homeostasis. Furthermore, the rest of this paragraph does not relate to the topic sentence making the purpose of the paragraph confusing. It seems to me the purpose of this paragraph was to talk about the ways in which NPC help regulate normal hepatic function while the subsequent paragraphs discuss how NPC, regulated by TREM1 and TREM2, regulate pathogenic processes.

Answer: We have deleted the redundancy and re-summarized the functions of non-parenchymal cells in ‘liver cell type’ section. In Figure 3, the function of TREM1 and TREM2 in liver pathological changes is shown.

Comment 5: Figure 1: if TREM2 negatively regulates LPS signaling, why are arrows pointing to MAPK and ERK? One would anticipate that a symbol indicating a blockade would be more appropriate. Also, the figure could be improved by including TREM1 signaling so that the reader can compare and contrast the effects of TREM1 and TREM2 on TLR4 signaling. Finally, the authors need to make is much more clear how TREM2 inhibits TLR4 signaling – it is counterintuitive that kinase activation would inhibit TLR4-mediated signaling. If the precise mechanism by which TREM2 negatively regulates TLR4 signaling is unknown, then state this, otherwise, provide the mechanistic data to clarify this important point. For example, and based on a information suggesting that low avidity binding of ligands to TMEM2 induces SHP phosphatase, is this a/the mechanism by which TREM2 can negatively regulate TLR4 signaling (reference 84)?

Answer: We have revised Figure 1 and added information about TREM1 in inflammation. The negative regulation of TREM2 in inflammation is dependent on the partial phosphorylation of ITAM, then the downstream signaling pathway is inhibited. The inhibitor A20 is also activated by partial phosphorylation of ITAM and inhibits NF-κB mediated inflammation.

Comment 6: Beginning in line 218 (TREM in metabolism) there is considerable data regarding now macrophages regulate adipose tissue. Why do the authors only focus on NK cells? A more balanced approach would be to include both macrophages and NK cells. If there is no information on TREM1 or 2 in adipose, then state this is an excellent area for future investigation.

Answer: In adipose tissue, TREM1 and TREM2 is not well investigated. Current research indicates that TREM1 and TREM2 mainly express on NK cells, KCs and macrophage. Related statement is added in line 220-224.

Comment 7: There is a lot of discussion of TREM1 and TREM2 in non-hepatic organs or diseases which occur in the sections describing TREM1 and TREM2 in liver. All discussion of non-hepatic TREM function should be described before diving into liver-specific TREM function (i.e. introduction)

Answer: We have revised this problem and correct this section.

Comment 8: In the fibrosis section, please remove redundancy and explain fibrogenesis in a logical manner. For example, inflammation usually precedes fibrosis, yet inflammation is mentioned in the last sentence of this section’s introductory paragraph.

Answer: We have rewritten this section in line 245-251.

Comment 9: Figure 2 needs additional labels and or legends. What are the little colored bars supposed to represent? The legend needs revision for English grammar and usage.

Answer: We have revised Figure 2

Comment 10: Beginning in line 310: this paragraph needs to be re-written for clarity. The thoughts need to be reorganized to tell a more logical story. For example, it would make more sense to describe immune impairments, then mechanism, then how TREM is involved.

Answer: We have added related information in line 279-304. Chronic inflammation induces excessive proliferation of hepatocytes. TREM1 promotes inflammation, proliferation and migration. TREM2 is much more complex, it decreases inflammation and oxidative stress.

Minor comments

Comment 11: Abstract starting line 15: The authors state non-parenchymal cells (NPC) but then list macrophages and neutrophils which is somewhat confusing. For example, Kupffer cells (KC) are resident hepatic macrophages. This sentence could be improved by saying “…NPC and cells which infiltrate the liver in response to injury including monocyte-derived macrophages and neutrophils.” (or equivalent)

Answer: We have rewritten this sentence in line 16-18.

Comment 12: Abstract starting line 17: I think this sentence on TREM1 and TREM2 would be improved if you indicate that both receptors depend on TLR4 as stated in the review.

Answer: We have rewritten this sentence in line 18-21.

Comment 13: Line 38: correct the spelling of ‘lipopolysaccharide’

Answer: We have corrected this word.

Comment 14: Line 88: it is more typical to call the specialized hepatic vessels ‘sinusoids’ instead of ‘sinuses’. Also, here it would be informative to state the location at which this blood enters the liver.

Answer: We have changed this word and added more information about sinusoids in line 77-79.

Comment 15: Line 109: somewhere in the discussion of HSC, the authors should state ‘perisinusoidal space of Disse’

Answer: We have added information about perisinusoidal space of Disse in line 101-102.

Comment 16: Line 126: You should state something like ‘infiltrating macrophages’ or ‘moncyte-derived macrophages’ to distinguish these macrophages from KC which are also macrophages.

Answer: We have changed the word into monocyte-derived macrophages in line 109-110.

Comment 17: Beginning line 131: how does upregulation of TREM1 increase macrophage and neutrophil recruitment? You state TLR4, but, presumably, this is mediated by macrophage and neutrophil-tropic chemokines, respectively, induced through TLR4 signaling. Can you be more specific behind the mechanisms involved?

Answer: In the new version, this sentence has been deleted. Although the mentioned that TREM1 recruit macrophage and neutrophil, the detailed mechanism is not clear.

Comment 18: Line 154: do you mean ‘because’ instead of ‘although’ to start this sentence?

Answer: We have changed this word

Comment 19: Line 251: please include a reference for the last sentence in this paragraph.

Answer: This sentence is a summary. The related references have been cited before.

Comment 20: Line 288: note that acute inflammatory responses can also cause hepatocyte death.

Answer: We have deleted the sentence.

Comment 21: Line 283: instead of ‘fibrosis’ do you mean ‘fibrogenic potential’ of HSC?

Answer: We have changed ‘fibrosis’ into ‘fibrogenic potential’

Comment 22: Line 312: do you have a reference backing this assertion?

Answer: The citation has been added in this sentence.

Comment 23: The legend to figure 3 needs more explanation. Moreover, it might be relevant to think about adding blocking arrows to show how TREM1 and TREM2 counterbalance one another.

Answer: The Figure 3 has been revised. The balance between TREM1 and TREM2 is time-dependent.

Comment 24: Line 367: tumor progression (and not ‘progress’?)

Answer: We have changed this word.

Reviewer 3 Report

The review article presented by Huifang Sun et al. clearly summarize and discuss recent published data on the function of the triggering receptors expressed on myeloid cells (TREM) 1 and 2 in liver injury, nonalcoholic steatohepatitis (NASH), hepatic fibrosis, and hepatocellular carcinoma (HCC) based on inflammatory regulation. TREM1 and TREM2 perform opposite functions: TREM1 plays a disease-promoting role via amplifying inflammation, promoting lipid accumulation, hepatic fibrosis, and accelerating tumor progress. TREM2 plays a protective role by suppressing inflammation, decreasing lipid-related gene expression, and suppressing tumor progress.

The Authors first focus on the liver cell types and their surface expression of TREM1 and TREM2 and then illustrate the different pathways of these two receptor to exercise their function in metabolism, hepatic fibrosis and liver tumorigenesis.

The Authors conclude that the recent understandings about the mechanisms through which TREM regulate inflammation, and cancer in liver, might provide new therapeutic insight for future treatment of these diseases through modulation of TREM1 and TREM2 expression

The review include a balanced, comprehensive and critical view of the research area.

The present version of the review is well written.

Minor points:

  1. -The Authors must carefully review references section. A lot of information is missing:
  • Minor points:

    1. -The Authors must carefully review references section. A lot of information is missing:
    • Ref. 7, 23, 73, 74, 75, 80, 83: the number of page is missing
    • Ref. 53: the last page number is missing
    • Ref. 36, 98: the number of volume and page is missing
    • 26, 62, 63, 66, 67, 69, 94, 116, 119, 120, 121: the name of the journal is missing
    • Ref. 66, 119: control the name of the Authors
    • The authors should uniform how write the name of journals; sometimes they are written in extenso and sometimes are abbreviated. For example see ref. 1 and ref. 60
    1. Line 293: The number of the paragraph is 3 and not 4.
    2. Explain the meaning of NASH in the legend to fig. 3
    3. Correct some typos errors

Author Response

Response to Reviewer 3:

Minor comments

Comment 1: The Authors must carefully review references section. A lot of information is missing:

Ref. 7, 23, 73, 74, 75, 80, 83: the number of page is missing

Ref. 53: the last page number is missing

Ref. 36, 98: the number of volume and page is missing

26, 62, 63, 66, 67, 69, 94, 116, 119, 120, 121: the name of the journal is missing

Ref. 66, 119: control the name of the Authors

The authors should uniform how write the name of journals; sometimes they are written in extenso and sometimes are abbreviated. For example see ref. 1 and ref. 60

Answer: We have revised the reference and correct wrong format

Comment 2: Line 293: The number of the paragraph is 3 and not 4.

Answer: We have correct the number.

Comment 3: Explain the meaning of NASH in the legend to fig. 3

Answer: The Figure 3 has been revised.

Round 2

Reviewer 1 Report

I do not think that this review paper should be accepted in this journal. There are still many statements that are incorrect. Figure 2 and 3 are completely incorrect and the message there is not clear.

Just as an example, what do they mean in Figure Legend 2 with "The decreased expression of TREM1 in KCs reduces cytokine and weakens the activation of quiet hepatic stellate cells. In the context of low inflammatory level, activated hepatic stellate cells reverse into quiet state and apoptosis."

What do authors mean whit quiet state? Do they mean quiescent?

Similarly Figure 3 is completely wrong, TREM1 and TREM2 play a cell-type specific role in different contexts.... what do authors mean with TREM1+ cells and TREM2+ cells?

The abovementioned examples include just two examples again of inaccurate sentences and misleading figures but there are many more all along this manuscript.

Authors have not substantially addressed our previous concerns in any way whatsoever.

Author Response

Comment 1: Extensive editing of English language and style required.

Answer: Before the manuscript was submitted, we had re-edited the manuscript through employing LetPub (https://www.letpub.com/) for its linguistic assistance. During the revision process, we have modified English language again.

Comment 2: I do not think that this review paper should be accepted in this journal. There are still many statements that are incorrect. Figure 2 and 3 are completely incorrect and the message there is not clear. Just as an example, what do they mean in Figure Legend 2 with "The decreased expression of TREM1 in KCs reduces cytokine and weakens the activation of quiet hepatic stellate cells. In the context of low inflammatory level, activated hepatic stellate cells reverse into quiet state and apoptosis." What do authors mean whit quiet state? Do they mean quiescent? Similarly Figure 3 is completely wrong, TREM1 and TREM2 play a cell-type specific role in different contexts.... what do authors mean with TREM1+ cells and TREM2+ cells? The above mentioned examples include just two examples again of inaccurate sentences and misleading figures but there are many more all along this manuscript. Authors have not substantially addressed our previous concerns in any way whatsoever.

Answer: Thank you very much for your comments. In Figure legend 2, “quiet state” means “quiescent state”. The word "quiescent" is used correctly in the text. It is our negligence to use the informal word "quiet" in the figure legend 2 and we have corrected “quiet” to “quiescent”. In Figure 3, TREM1+ cells and TREM2+ cells are cells who express TREM1 and TREM2. “TREM1+” and “TREM2+”have been used in many papers (Ref 1, 2). TREM1 and TREM2 are expressed in different liver related cells, these cells are involved in inflammation, metabolism, fibrosis and tumorigenesis. In Figure 3, we used TREM1+/TREM2+ cells to represent different cells expressed TREM1/TREM2. Then, we further listed which kind of cell was affected in different biological process, and what biological changes have taken place. To make the Figure 3 easier to understand, we have changed “TREM1+/TREM2+ cells” into “TREM1/TREM2” and added defined cells in TREM1/TREM2 mediated biological regulation. Furthermore, we have revised the manuscript and rewritten some sentences that may be not stated clearly.

Reviewer 2 Report

The authors have adequately addressed my concerns.

Author Response

Comment 1: The authors have adequately addressed my concerns.

Answer: Thank you very much for your comments.

This manuscript is a resubmission of an earlier submission. The following is a list of the peer review reports and author responses from that submission.

Round 1

Reviewer 1 Report

This review summarizes recent research finding of Trem1 and Trem2 expression in the context of inflammation and metabolism and how liver pathologies could be related to the expression balance between these TREM receptors. While the authors do a reasonable attempt at presenting recent research findings, the review needs some polishing and grammar corrections. The authors do a good job at suggesting a possible link between TREM2 and liver pathologies, however, clearer links between the different types of cells within the liver are needed to make this conclusion. The authors need to incorporate recent scRNAseq data and discuss this at length, rather than presenting weaker findings to make their points. This is a timely review of an important topic but the authors will need to address all these points and revise their current manuscript in order to make their arguments clearer and increase the value of their work.

Major issues:

  • Although the subject of this review is needed in the field, this manuscript is poorly written. It has synthax and grammar issues throughout. It will need extensive revision to be understandable. This, in turn, presented difficulties in this reviewers’ review.
  • The description of structural component of TREM receptors needs to be verified in the introduction.
  • This claim needs citations: line 62: ‘they are also expressed on liver resident immune cells, such as Kuffer cells (KCs), Hepatic stellate cells 63 (HSCs), liver sinusoidal endothelial cells (LSECs), etc’.
  • Authors do not fully explain how they go from apparent Trem1/Trem2 expression in several liver cells (which is not referenced appropriately here) to regulation of inflammation. Saying (line 68): ‘From the above research results, we can see that the change of TREM1 and TREM2 expression is often associated with the development of liver inflammation’ is insufficient since this manuscript is meant to be a review not original research.
  • Inflammation doesn’t just ‘disappear with antigen removal’. The authors need to elaborate more on the role of wound healing/resolution mechanism in the context of chronic inflammation.
  • ‘Once pro-inflammation of TREM1 is over-activated, TREM2 will alleviate inflammatory response via TREM2-TLR4 pathway. These all show that TREM2 plays a protective role in innate immunity of liver’. The authors are contradicting themselves here since they cite earlier that LPS downregulates Trem2, so it is unlikely it opposes Trem1 in the same cell under these conditions and they do not reference a study where this is the case in different cells within the liver.
  • Authors use ‘show’ and ‘suggest’ interchangeably. Please be mindful of this. Most ‘show’ usage here should be replaced with ‘suggest’.
  • Authors should discuss the specific caveats of studies presenting Trem2 expression outside the myeloid compartment.

Minor issues:

  • Authors fail to connect section 2, where they attempt detailed review of liver structure, to the rest of the manuscript. At times it seems the sections where written by different authors and are not connected.
  • It is still not clear if inflammation precedes metabolism changes so this should be clarified.
  • Microglia exist only in the brain so the term ‘microglia of the brain’ is unnecessary.
  • Line 51: ‘With the upregulation of TREM1, the expression of TREM2 is downregulated in acute liver injury’ needs citation.
  • The point that ‘TLR4 is highly related to TREM expression’ is made several times throughout the manuscript. I strongly encourage revision of this.   It is repetitive and only one reference for mechanical ventilation injury is cited for this claim.  There needs to be more support for this claim

Reviewer 2 Report

Reviewer Comments to Author:

This manuscript is a review that aims to describe the role of TREM1 and TREM2 in liver disease. Although the manuscript could contain some interesting data there are important deficiencies, including lack of accuracy and important mistakes when describing the results that are published as well as missing information regarding basic concepts of the signalling pathway and many grammatical and conceptual errors all along the manuscript.

Major Comments

  1. All along the manuscript there are a lot of vague sentences that are unspecific and incorrect therefore being unacceptable from the scientific point of view.

Below are included some of the many examples lacking any sense;

Page 2. “Generally, TREM1 and TREM2 are expressed in most immune cells and they work through inflammation.”

Page 2. “ There are many classifications of liver diseases, but most of them are chronic disease, such as liver injury, fatty liver, hepatic fibrosis and liver cancer.”

Page 3. “In liver TREM1 and TREM2 are the same.”

Page 3. “Comparing with TREM2 expression in LPS induced acute injury, expression of TREM2 is upregulated in non-parenchymal liver cells of chronic liver injury.”

Page 4. “The signal passed from TREM2 to DAP12 is negatively regulated. Furthermore, the Syk-mediated MAPK signal pathway is dampened too.”

Page 5. “Pathogen of Nonalcoholic fatty liver is much complex.”

Page 6. “In connective tissue, the direct reason of fibrosis is the improvement of ECM level.”

These include just some examples but many of the sentences in this manuscript lack accuracy, are inappropriate and even incongruous.

  1. Some of the sentences are even unacceptable.

Page 3. “When HSC is activated, it promotes the fibrotic process, such as the synthesis of vinegar sperm extracellular matrix.”

  1. There are many grammatical mistakes in many of the sections.

In general, this paper does not meet the quality that is characteristic of this journal and will modestly increase our understanding of the role of TREM1 and TREM2 in liver diseases. 

Reviewer 3 Report

TREM family receptor biology is an emerging and exciting research area across organ systems. Sun et al. sought to provide a comprehensive review on what is known regarding the biology of two TREM receptors, TREM1 and TREM2, in liver-related diseases. This is a timely review given the large number of peer reviewed studies exploring TREM1 and TREM2 biology in liver biology. With consideration of the following points, this manuscript would be a wonderful addition to the peer reviewed literature on this topic.

  1. First and foremost, the manuscript must be edited for English grammar and usage. There are too many examples to note them all, but for example, ‘researches’ is not appropriate as used throughout the manuscript.
  2. The authors need to do a much better job of referencing research in this field. While I was reviewing the literature, I noticed that many very relevant and impactful studies were not mentioned
  3. There are several places where statements are made which were not supported with a reference.
  4. The relationship between LPS and TREM must be made very clear (at present, it is not). Is LPS a bona fide ligand of TREM1 or TREM2?
  5. There are also several inaccuracies found within the manuscript. For example, around line 100 when discussing the origins of Kupffer cells, the authors state that they arise from monocytes. There is now extensive lineage tracing literature which demonstrates that this is not entirely true. Also, there are papers which show HSC do not express TREM2. The authors must mention this discrepancy. Finally, while adipose inflammation is clearly a critical regulator of liver disease, it is not solely dependent on NK cells as the authors intimate around line 216. Macrophages are major contributors and are not mentioned.
  6. There is also significant redundancy in this manuscript. For example, information on the general biology of TREM1 and TREM2 should be found only in the intro, and not other places in the manuscript where TREM1 and TREM2 are mentioned.
  7. Regarding alcohol-induced liver disease, the authors should change their nomenclature to reflect thinking in the field. Specifically, ‘alcoholic liver disease’ should be changed to ‘alcohol-associated liver disease’. Moreover, when alcohol-associated liver disease is mentioned, a key paper exploring the role of TREM1 is not mentioned. Finally, although alcohol-associated liver disease can cause metabolic disturbances, it is not clear why that is included in a discussion which otherwise should focus on obesity and metabolic syndrome. It just seems a bit out of place as currently described.
  8. Line 251: what is meant by a ‘fibrous protein’? Typically, type I and type III collagen are considered ‘fibrillar’ collagens, so this nomenclature seems a bit redundant. I suggest using a specific example.
  9. Table 1 is completely out of place in a manuscript focused on TREM1 and TREM2 in liver disease. A useful table would summarize the available literature profiling the role of these two receptors in liver disease as a companion to the text description of those papers.
  10. Line 324?? Not clear what the authors are trying to say here.
  11. Line 80 ‘cell construction of the liver’ would be better stated ‘liver cell types’ or equivalent.
  12. Line 107. What is ‘vinegar sperm extracellular matrix’?